# OpenReview forum: "Dynamic Post-Hoc Neural Ensemblers"
_ICLR.cc/2025/Conference — ICLR 2025 Conference Withdrawn Submission_

### Official Review · Reviewer_eck1 · 2024-10-30

**Soundness:** 3
**Presentation:** 3
**Contribution:** 2
**Rating:** 5
**Confidence:** 2

**Summary:**

The paper introduces a novel approach to post-hoc ensembling using neural networks, termed "Neural Ensemblers." These ensemblers dynamically generate weights for each base model in the ensemble on a per-instance basis, addressing the limitations of static ensembling methods. The authors propose a regularization technique inspired by DropOut in deep learning to mitigate the risk of overfitting and diversity collapse. The method is evaluated across various data modalities, including tabular data, computer vision, and natural language processing, demonstrating competitive performance against strong baselines.

**Strengths:**

1) The introduction of Neural Ensemblers as a dynamic post-hoc ensembling method is a significant contribution to the field. The use of neural networks to dynamically assign weights to base models on a per-instance basis is a novel and effective approach.

2) The proposed regularization technique, which involves randomly dropping base model predictions during training, is theoretically sound and empirically validated. This technique effectively mitigates the risk of overfitting and enhances ensemble diversity.

**Weaknesses:**

While the paper mentions the use of a fixed set of hyperparameters across all experiments, it does not explore the sensitivity of the Neural Ensemblers to hyperparameter changes. A more detailed analysis of hyperparameter tuning could provide deeper insights.

Does the author carefully tune the hyper-parameters of other baselines e.g., XGBoot?

The author should also compare their method with other advanced methods, e.g., LightGBM, CatBoot, which can be directly used in the open source libraries.

Lack of references and comparisons to Ensemble NN [1] and SCARF [2]. Both of these papers explore the ensemble or pretraining in Deep neural networks on tabular datasets.

[1] Zhang S, Liu M, Yan J. The diversified ensemble neural network[J]. Advances in Neural Information Processing Systems, 2020, 33: 16001-16011.

[2] Bahri D, Jiang H, Tay Y, et al. Scarf: Self-Supervised Contrastive Learning using Random Feature Corruption[C]//International Conference on Learning Representations.

**Questions:**

See weaknesses.

---

> ### Author Response · Authors · 2024-11-21
> **Response to reviewer eck1**
>
> Dear reviewer,
> thank you for your feedback and interesting insights. Below we answer your questions:
>
> **Weaknesses**
>
> 1. ***While the paper mentions using a fixed set of hyperparameters across all experiments, it does not explore the sensitivity of the Neural Ensemblers to hyperparameter changes. A more detailed analysis of hyperparameter tuning could provide deeper insights.***
>
> We appreciate your emphasis on the importance of hyperparameter analysis. In response, we conducted an ablation study on the number of layers and the hidden dimension of our Neural Ensembler. The results, presented in **Appendix J (Figure 13)** of the updated manuscript (or [here](https://anonymous.4open.science/r/NeuralEnsemblers/anonymous_rebuttal/figure13.png), demonstrate that our default configuration strikes a balance that provides robust results across all the metadatasets.
>
> 2. ***Does the author carefully tune the hyper-parameters of other baselines e.g., XGBoost?***
>
> We used the default settings recommended in the respective libraries. These methods are well-researched with established default hyperparameters. Similarly, we kept the hyperparameters of our method fixed throughout the experiments to ensure a fair comparison.
>
> 3. ***The author should also compare their method with other advanced methods, e.g., LightGBM, CatBoot, which can be directly used in the open source libraries.***
>
> We included additional baselines such as *CatBoost* and  *XGBoost* in **Appendix I (Tables 18 and 19)** of the updated manuscript (also available [here](https://anonymous.4open.science/r/NeuralEnsemblers/anonymous_rebuttal/table19.png) and [here](https://anonymous.4open.science/r/NeuralEnsemblers/anonymous_rebuttal/table20.png)). We did not include LightGBM as it is similar to the Gradient Boosting baseline already evaluated.
>
> 4. ***Lack of references and comparisons to Ensemble NN [1] and SCARF [2]. Both of these papers explore the ensemble or pretraining in Deep neural networks on tabular datasets.***
>
> Thank you for bringing these references to our attention. Unfortunately, we cannot find an open-sourced implementation of Ensemble NN to include it as a baseline. However, we discussed it in the related work, **Section 5**, of the updated manuscript. While SCARF is indeed a very interesting approach, it focuses on pretraining for tabular data rather than ensembling, thus falling outside the primary scope of our study.
>
>
> ---
>
> Finally, thank you very much for taking the time to review our paper and providing us with valuable feedback, prompting several improvements to our submission. We hope we have clarified some of your concerns and would appreciate it if you consider improving the rating in your review. We welcome any further suggestions or feedback that you would be willing to share.

---

> > ### Comment · Reviewer_eck1 · 2024-11-22
> > **Further questions**
> >
> > Thank you for the author's response and additional results, which has alleviated some of my concerns. However, I still want to see the results of LightGBM, which is more efficient than XGBoost and CatBoost. Due to its proposed One-side sampling and EFB (Exclusive Feature Bundling), LightGBM often achieves better results on tabular data. This has made LightGBM a frequent choice in many data science competitions.

---

> ### Author Response · Authors · 2024-11-23
> **Response to further questions**
>
> Thank you for your continued engagement and for acknowledging the additional results we provided.
>
> To address it, we run *LightGBM* and compare it against all baselines. See results in Table 18 and 19 from the submitted PDF or [here](https://anonymous.4open.science/r/NeuralEnsemblers/anonymous_rebuttal/table18.png) and [here](https://anonymous.4open.science/r/NeuralEnsemblers/anonymous_rebuttal/table19.png). Our results are consistent with previous works that have shown CatBoost and XGBoost outperforms LightGBM across various datasets [1, 2, 3].
>
> We appreciate your feedback and hope our response addresses your concerns which would reflect in the score increase.
>
> References:
>
> [1] Kohli R., Feurer M., Eggensperger K., Bischl B., Hutter F. (2024). Towards Quantifying the Effect of Datasets for Benchmarking: A Look at Tabular Machine Learning. ICLR 2024 Workshop on Data-centric Machine Learning Research.
>
> [2] Salinas, D.,  Erickson, N. (2024). TabRepo: A Large Scale Repository of Tabular Model Evaluations and its AutoML Applications. AutoML Conference 2024 (ABCD Track).
>
> [3] McElfresh, D., et al. (2023). When Do Neural Nets Outperform Boosted Trees on Tabular Data? Advances in Neural Information Processing Systems.

---

### Official Review · Reviewer_62Pt · 2024-11-03

**Soundness:** 2
**Presentation:** 1
**Contribution:** 2
**Rating:** 3
**Confidence:** 3

**Summary:**

A common approach to combining predictions from multiple ensemble members is to assign equal weights to each prediction, as seen in hard and soft voting methods. However, it is clear that certain ensemble members may offer better predictions for specific instances, and it leads to the concept of weighted voting. This work proposes obtaining such instance-wise ensemble weights through a separate neural network, referred to as the Neural Ensembler.

**Strengths:**

1. The problem statement is well-motivated; it would be beneficial to adaptively determine ensemble weights for a given instance.

2. The authors propose two setups: "stacking" and "model-averaging." In the stacking setup, the Neural Ensembler directly outputs the ensembled prediction. In contrast, the model-averaging setup has the Neural Ensembler provide the weights for each ensemble member. While the stacking setup aligns with the ultimate goal of obtaining an ensembled prediction, the model-averaging setup is also valuable as it explicitly reveals the contribution of each ensemble member.

**Weaknesses:**

1. There are concerns about fair comparison since the proposed Neural Ensemble ultimately learns from additional validation data. While baseline methods also train statistical models based on validation data, the issue is that the Neural Ensemble is significantly larger than the baseline methods. For example, methods like Greedy, Quick, and BO train statistical models with $M$ (unnormalized) ensemble weights as parameters. In contrast, NE-Stack trains a statistical model (neural network) with $D \times 32 + 32 \times 32 + 32 \times 32 + 32 \times D$ parameters, which amounts to $2688$, $8448$, and $66048$ parameters for $D=10$, $100$, and $1000$, respectively. Given the significant difference in the scale of the introduced statistical models, ranging from hundreds to tens of thousands of times larger, it is questionable whether introducing the same additional validation data could make a fair comparison.

2. I believe the assumption of having a sufficiently large amount of validation data, which can be used to train the neural network model, is quite strong. If the "Avg. samples for Validation" shown in Table 1 indicates the size of the validation dataset used to train the Neural Ensembler and the statistical models of the baseline methods for each dataset, it appears particularly large for NASBench and FTC. Conducting an ablation study on this would be valuable to assess how much validation data is necessary for the Neural Ensembler to be effective. Currently, the only ablation study presented is on the dropout probability, aimed at preventing overfitting on the given validation data.

3. Additionally, the Neural Ensembler is implemented using an MLP architecture with a hidden dimension of 32 and a depth of four. Such architectural choice also warrants an ablation study; a model that is too large may incur significant costs and increase overfitting, while one that is too small may not perform adequately. Currently, there is no discussion regarding the size of the Neural Ensembler model.

4. The key feature of the proposed Neural Ensembler is its ability to adjust the weights of each ensemble member in an instance-specific manner, either implicitly (in a stacking setup) or explicitly (in a model-averaging setup). However, there is currently no direct evidence regarding its effectiveness from this perspective. One potential approach could involve directly learning the weighting coefficients for each instance (i.e., oracle) and evaluating how closely the ensembled predictions (or ensemble weights in the model-averaging setup) generated by the Neural Ensembler align with them.

5. Above all, are the values reported in the tables the averages and standard deviations of the results for the sub-datasets within each meta dataset? The fact that nearly all results overlap within the range of standard deviations suggests that the experimental results may not be statistically significant. A clear clarification on this matter seems necessary.

**Questions:**

1. I think the assumption of having a sufficiently large amount of validation data (that can be used to train the neural network model) is quite strong. In such cases, further exploration of existing methodologies based on well-grounded principles (such as Bayesian principles) would be valuable.

2. Specifically, one can approximate the model probability in Bayesian model averaging using the Bayesian Information Criterion (BIC), where the likelihood is estimated from the validation set.

3. Following BIC, the harmonic mean estimator and its variants come to mind. A recent relevant work is "Machine Learning Assisted Bayesian Model Comparison: Learnt Harmonic Mean Estimator" by McEwen et al. (2023). Have you explored such classical methods grounded in Bayesian principles alongside BO-based approaches?

4. I recognize the value of presenting normalized metrics for relative comparisons in the main text. However, it would be helpful to include the raw numbers in the appendix. Relying solely on relative metrics can make it challenging to assess the significance of the results for each methodology.

---

> ### Author Response · Authors · 2024-11-21
> **Response to reviewer 62Pt**
>
> Dear reviewer,
> thank you for your valuable feedback and insightful comments. We have addressed your concerns below:
>
> **Weaknesses**
>
> 1. ***There are concerns about fair comparison since the proposed Neural Ensemble ultimately learns from additional validation data. While baseline methods also train statistical models based on validation data, the issue is that the Neural Ensemble is significantly larger than the baseline method. [...]***
>
> We understand your concern regarding the use of additional validation data and the size of the Neural Ensembler. **It is important to note that all methods in our comparison, except for the *Random* baseline, use validation data. For instance, Greedy approach uses the validation data for selecting iteratively the model that decreases the loss if added to the ensemble. SVM and RF  uses the validation data to optimize the respective model parameters.
>
> In fact, some baselines overfit the validation data, as observed by comparing **Figures 4 and 8** in our manuscript, which means they are expressive enough, and in turn, that this is not an issue of lack of expressivity or underparametrization of the baselines.
> To address the concern about Neural Ensembler’s dependence on validation data, we conducted an ablation study varying the percentage of validation data used for its training. The results, presented in **Appendix K (Figure 14)** of the updated manuscript (or [here](https://anonymous.4open.science/r/NeuralEnsemblers/anonymous_rebuttal/pct_valid_data.pdf)), demonstrate that our method remains robust when the amount of validation data is significantly reduced. This indicated that our method does not heavily depend on a large validation set.
>
> 2. ***I believe the assumption of having a sufficiently large amount of validation data, which can be used to train the neural network model, is quite strong. [...]***
>
> Thank you for highlighting this point. As mentioned, our ablation study in **Appendix K (Figure 14)** (or  [here](https://anonymous.4open.science/r/NeuralEnsemblers/anonymous_rebuttal/pct_valid_data.pdf)) shows that our Neural Ensembler performs well even with limited validation data.
>
> 3. ***Additionally, the Neural Ensembler is implemented using an MLP architecture with a hidden dimension of 32 and a depth of four. Such architectural choice also warrants an ablation study; a model that is too large may incur significant costs and increase overfitting, while one that is too small may not perform adequately. [...]***
>
> We appreciate your suggestion regarding the architectural choices of the Neural Ensembler. In response, we conducted an ablation study varying the number of layers and the hidden dimension of the MLP architecture. The results are presented in **Appendix J (Figure 13)** of the updated manuscript (or [here](https://anonymous.4open.science/r/NeuralEnsemblers/anonymous_rebuttal/figure13.png)). Our results show that the original architecture strikes a balance between performance and complexity. While modifying the architecture can lead to performance improvements on certain datasets, it decreases the performance on others. This demonstrates that our chosen configuration is robust across diverse datasets, providing consistent results without the need for extensive hyperparameter tuning.

---

> > ### Comment · Reviewer_62Pt · 2024-11-22
> >
> > > This demonstrates that our chosen configuration is robust across diverse datasets, providing consistent results without the need for extensive hyperparameter tuning.
> >
> > The logic here remains unclear. Ablation studies indicate that deviating from the originally presented configuration, by modifying the number of hidden layers or hidden dimensions, results in significant performance changes. It is unclear how this aligns with the claim that the method works _"without the need for extensive hyperparameter tuning"_?
> >
> > Fundamentally, how was the original configuration of $L=4$ and $H=32$ determined? Are you suggesting that this setup, chosen arbitrarily without extensive tuning, just happened to deliver the best performance? If $L=4$ and $H=32$ were specifically tailored for the presented experiments, it raises concerns about the fairness of comparisons with other baselines, which relied on default hyperparameters (as noted in the response to Reviewer eck1)

---

> ### Author Response · Authors · 2024-11-21
> **Response to reviewer 62Pt [continuation]**
>
> 4. ***The key feature of the proposed Neural Ensembler is its ability to adjust the weights of each ensemble member in an instance-specific manner, either implicitly (in a stacking setup) or explicitly (in a model-averaging setup). However, there is currently no direct evidence regarding its effectiveness from this perspective. [...]***
>
> We kindly want to point out that our proof-of-concept experiment, detailed in our paper **(Figure 3)**, demonstrates the advantage of dynamic ensembling in a one-dimensional toy problem. Additionally, our extensive empirical results in a broad range of modalities and datasets provide evidence of the effectiveness of instance-specific weighting. While we value the experiment that you suggested to test this hypothesis, we believe that learning a Neural Ensembler that could approximate the oracle would be challenging, as there could be many possible ensemble weights that also fit the data (besides the oracle). In other words, we might learn a set of weights, different from the oracle, that also fits the data.
>
> In our experiments, we consider ensembles with constant weights that do not depend on the instance (see table 1 and 2, baseline MA), which lean fixed weights so that the output is $y=\theta_1 z_1 + …\theta_n  z_n$, where z are the model predictions. As we can see in these tables, learning dynamically the ensemble (NE-MA method) outperforms the fixed weight approach MA.
>
> 5. ***The fact that nearly all results overlap within the range of standard deviations suggests that the experimental results may not be statistically significant. [...]***
>
> We appreciate your observation regarding statistical significance. To understand the significance of the results, we included Critical Difference (CD) diagrams in **Appendix H** of the updated manuscript. These diagrams (**Figure 11h** or [here](https://anonymous.4open.science/r/NeuralEnsemblers/anonymous_rebuttal/CD-Difference.pdf)) show that our method achieves top performance across multiple datasets and performs best when aggregating all datasets. There is no baseline better than our approach, and, given the ablation studies, it still has room for improvement.
>
> We additionally found out that our method is *significantly* better than *Greedy* (the second best approach) in the metadatasets with a lot of number of classes (**appendix N**). We hypothesize this occurs because of the neural ensembler's design, as it shares parameters across different classes (section 3.1).
>
>  Moreover, our method is faster than the *Greedy* baseline (**Figure 9**), which is the second-best method. We achieved these results using the same Neural Ensembler configuration across all metadatasets, highlighting its robustness. We could further improve the performance per metadataset by carefully tuning the hyperparameters (**Appendix J (Figure 13)** or [here](https://anonymous.4open.science/r/NeuralEnsemblers/anonymous_rebuttal/figure13.png)).
>
> We also emphasize the theoretical contribution of our work, including proofs that our method provides a lower bound on ensemble diversity, which is crucial to avoid overfitting during ensemble learning.
> For further details, please refer to the response to reviewer Svxb (first weakness point).
>
> **Questions**
>
> 1. ***I think the assumption of having a sufficiently large amount of validation data [...] is quite strong [..]***.
>
> As previously mentioned, we do not use assume the availability of a large validation set. Our ablation study in **Appendix K (Figure 14)** (or  [here](https://anonymous.4open.science/r/NeuralEnsemblers/anonymous_rebuttal/pct_valid_data.pdf) demonstrates that our method maintains strong performance even with significantly reduced validation data.
> For further details, please refer to the response to reviewer Svxb (second question).
>
> 2-3. ***One can approximate the model probability in Bayesian model averaging using the Bayesian Information Criterion (BIC), where the likelihood is estimated from the validation set. [...] Following BIC, the harmonic mean estimator and its variants come to mind.***
>
>
> Thank you for highlighting the relevance of Bayesian methods for increasing performance in the context of ensemble learning. In response, we would like to argue that our method approach uses a modified version of dropout, which can be interpreted as a Bayesian approximation, as discussed in [1]. We acknowledge that methods like BIC and harmonic mean estimators could potentially improve performance. To this end, we included a brief discussion of these methods in **Appendix B** of the updated manuscript.
>
> [1] Gal, Y., & Ghahramani, Z. (2016, June). Dropout as a bayesian approximation: Representing model uncertainty in deep learning. In international conference on machine learning (pp. 1050-1059). PMLR.

---

> > ### Comment · Reviewer_62Pt · 2024-11-22
> >
> > > Our ablation study in Appendix K (Figure 14) (or here demonstrates that our method maintains strong performance even with significantly reduced validation data.
> >
> > > Specifically, when using just 10% of the validation data, the normalized NLL remains below 1.25 for all metadatasets except TR-Class, indicating that the performance loss is less than 25% compared to using the full validation set. Even at 5% of validation data, the NE maintains reasonable performance levels.
> >
> > As I understand it, the Normalized NLL in Figure 14 uses the original results obtained with the full validation data as a baseline value of 1, with relative differences shown accordingly. However, considering that the performance gap between the proposed method and the baselines wasn’t particularly large to begin with, wouldn’t a result of 1.25 times the original indicate performance worse than the baselines? Perhaps it could be worth considering using a simple yet competitive baseline like Greedy (applied to a subset of the validation data) as the baseline for this ablation.
> >
> > Moreover, why do QT-Micro and QT-Mini exhibit worse performance at 20-40% compared to 10% (or even 5% and 1%)? Intuitively, performance should tend to degrade as the validation data decreases from 100% to 0%. Additionally, there are instances—presumably around 1%—where the Normalized NLL approaches 1 (e.g., NB, QT-Micro, QT-Mini), which also seems odd.

---

> > > ### Comment · Reviewer_62Pt · 2024-11-22
> > >
> > > While examining Table 3 (to see the baseline performance), I noticed something strange. How is it possible for Greedy, which achieves a value of <1 in NB (100), to achieve =1 in NB (1000)? When Greedy's performance is 1, it suggests that after selecting the initial single best candidate, all other candidates provide no improvement in validation performance when added to the ensemble, meaning the ensemble construction ends with just the single best candidate. How is it that, in NB (1000), none of the 999 other candidates contribute any ensemble gain? More importantly, the fact that Greedy achieves <1 in NB (100) implies that at least one of the 100 candidates improves the ensemble. Does this mean that in NB (100), there is, by chance, at least one candidate that improves the ensemble, but in NB (1000), there is no such candidate among the 1000?

---

> ### Author Response · Authors · 2024-11-21
> **Response to reviewer 62Pt [continuation 2]**
>
> 4. ***It would be helpful to include the raw numbers in the appendix.***
>
> We appreciate your suggestion for greater transparency. In response, we added tables with the unnormalized results in **Appendix E.1** of the updated manuscript (also available [here](https://anonymous.4open.science/r/NeuralEnsemblers/anonymous_rebuttal/table16.png) and [here](https://anonymous.4open.science/r/NeuralEnsemblers/anonymous_rebuttal/table17.png)).
>
> ---
>
> Finally, thank you very much for taking the time to review our paper and providing us with valuable feedback, prompting several improvements to our submission. We hope we have clarified some of your concerns and would appreciate it if you consider improving the rating in your review. We welcome any further suggestions or feedback that you would be willing to share.

---

> ### Author Response · Authors · 2024-11-23
> **Response to reviewer 62Pt**
>
> > ***The logic here remains unclear. Ablation studies indicate that deviating from the originally presented configuration, by modifying the number of hidden layers or hidden dimensions, results in significant performance changes. It is unclear how this aligns with the claim that the method works "without the need for extensive hyperparameter tuning"?***
>
> In the original paper version, we explain in Lines 352-254 that we find the hyperparameters (HPs) using a separate metadataset split from Quicktune which is not used for the final test. This set of HPs is *fixed* because they do not change across metadatasets, even though they are from different data modalities.
>
> > ***As I understand it, the Normalized NLL in Figure 14 uses the original results obtained with the full validation data as a baseline value of 1, with relative differences shown accordingly. However, considering that the performance gap between the proposed method and the baselines wasn’t particularly large to begin with, wouldn’t a result of 1.25 times the original indicate performance worse than the baselines? Perhaps it could be worth considering using a simple yet competitive baseline like Greedy (applied to a subset of the validation data) as the baseline for this ablation.***
>
> As we mention in the first bullet point from our response, all baselines except Random use validation data, so we expect that they also will have a drop in performance, as they will use less validation data.
>
> > ***Moreover, why do QT-Micro and QT-Mini exhibit worse performance at 20-40% compared to 10% (or even 5% and 1%)? Intuitively, performance should tend to degrade as the validation data decreases from 100% to 0%. Additionally, there are instances—presumably around 1%—where the Normalized NLL approaches 1 (e.g., NB, QT-Micro, QT-Mini), which also seems odd.***
>
> We thank the reviewer for raising this point. We realize a problem in the plotting for this rebuttal experiment, which we fixed in the updated version (see Figure 14 or [here](https://anonymous.4open.science/r/NeuralEnsemblers/anonymous_rebuttal/pct_valid_data.pdf)). Indeed, we can see that our method is affected by decreasing the validation data size.  We also revise the replies regarding the discussions around this plot.
> For some metadatasets the decrease is relatively small (FTC, NB), while for some is larger (QT-Micro). We hypothesize this occurs because the average number of samples in the validation split is already relatively small for QT-Micro (see Table 5).
>
> > ***While examining Table 3 (to see the baseline performance), I noticed something strange. How is it possible for Greedy, which achieves a value of <1 in NB (100), to achieve =1 in NB (1000)? When Greedy's performance is 1, it suggests that after selecting the initial single best candidate, all other candidates provide no improvement in validation performance when added to the ensemble, meaning the ensemble construction ends with just the single best candidate.***
>
> We invite the reviewer to bear in mind that our results are reported on a separate **test** data split. We will likely find an ensemble that improves the performance over a single model on the **validation** data split. However, it is possible that this ensemble is not generalizing well, thus having lower performance in the test split.
>
> Imagine a situation where you tune model hyperparameters using a validation split, and you find a good hyperparameter configuration on this split. It can happen that it does not translate to a test because we probably overfit the validation split. This behavior has been seen in HPO (hyperparameter optimization) [1][2], and it is a plausible situation here as well.
>
> [1] Lévesque, Julien-Charles. "Bayesian hyperparameter optimization: overfitting, ensembles and conditional spaces." (2018).
>
> [2] Makarova, Anastasia, et al. "Automatic termination for hyperparameter optimization." International Conference on Automated Machine Learning. PMLR, 2022.

---

> > ### Comment · Reviewer_62Pt · 2024-11-23
> >
> > > We thank the reviewer for raising this point. We realize a problem in the plotting for this rebuttal experiment, which we fixed in the updated version (see Figure 14 or here). Indeed, we can see that our method is affected by decreasing the validation data size. We also revise the replies regarding the discussions around this plot. For some metadatasets the decrease is relatively small (FTC, NB), while for some is larger (QT-Micro). We hypothesize this occurs because the average number of samples in the validation split is already relatively small for QT-Micro (see Table 5).
> >
> > Thank you for providing further clarification. If I may ask, could you share what might have contributed to the previous results? Was there possibly an issue with the experimental setup, or could other factors have influenced the outcome?
> >
> > > We invite the reviewer to bear in mind that our results are reported on a separate test data split. We will likely find an ensemble that improves the performance over a single model on the validation data split. However, it is possible that this ensemble is not generalizing well, thus having lower performance in the test split.
> >
> > I understand that the Greedy approach first constructs an ensemble using the validation split and then evaluates it on the test split. However, the fact that the evaluation result on the test split is exactly 1.000 suggests that a single-best model was indeed chosen during the ensemble construction stage using the validation split. In other words, we have not succeeded to find an ensemble that outperforms a single model on the validation split, contrary to the expectation expressed in the statement, *"We will likely find an ensemble that improves the performance over a single model on the validation data split. However, it is possible that this ensemble is not generalizing well, thus having lower performance in the test split."*

---

> > > ### Author Response · Authors · 2024-11-26
> > > **Reply to reviewer**
> > >
> > > Again, we thank the reviewer for pointing out this experiment. No issue related to the mentioned plot affected the whole experimental setup.
> > >
> > > Additionally, we thank the reviewer for the effort and the engaging discussion. We will incorporate the feedback in our future version.

---

> ### Author Response · Authors · 2024-11-23
> **New akaike weighting baseline added**
>
> As suggested by the reviewer, we included a baseline that considers akaike-weighting [1] or pseudo bayesian model averaging. We can see that the method obtains interesting results in some metadatasets such as *NB-Mini*, however it does not outperform our method. The reviewer is invited to check Table 18 and 19 from the updated PDF, or [here](https://anonymous.4open.science/r/NeuralEnsemblers/anonymous_rebuttal/table18.png) and [here](https://anonymous.4open.science/r/NeuralEnsemblers/anonymous_rebuttal/table19.png).
>
> [1] Wagenmakers, E. J., & Farrell, S. (2004). AIC model selection using Akaike weights. Psychonomic bulletin & review, 11, 192-196.

---

### Official Review · Reviewer_QKaE · 2024-11-04

**Soundness:** 3
**Presentation:** 3
**Contribution:** 2
**Rating:** 6
**Confidence:** 3

**Summary:**

The paper proposes Dynamic Post-Hoc Neural Ensemblers for replacing the traditional ensemble method. Their main contribution is the use of two MLPs to dynamically generate weights for ensemble members. Additionally, when training these MLPs, they randomly drop base models to reduce overfitting and improve generalization capabilities. Lastly, their empirical results demonstrate that Neural Ensemblers consistently form competitive ensembles across diverse data modalities, including tabular data (classification and regression), computer vision, and natural language processing.

**Strengths:**

**Innovative Approach:** The paper introduces Dynamic Post-Hoc Neural Ensemblers,  aiming to generate dynamic weights for ensemble members.

**Reduced Overfitting:** By randomly dropping base models during training, the method addresses overfitting and potentially improves generalization.

**Empirical Validation:** The paper provides empirical results showing the method’s outperformance and robustness across diverse datasets.

**Weaknesses:**

1. The study lacks consideration of computational efficiency. The Neural Ensemblers’ performance depends on a strong subset of 50 base models, selected by `DivBO`, which may increase computational requirements. The author should compare the Neural Ensembler's performance with that of a single model of equivalent size to better evaluate its efficiency. For example,  to evaluate Neural Ensember, the author could provide some specific efficiency metric comparisons, such as training time, inference time, or memory usage compared to baseline methods, or analyze the computational complexity of their approach.

2. My review of the author's code confirmed that the meta-datasets are limited to a tabular format, making the paper's description somewhat vague. A clearer explanation of the data structure would have been beneficial. Otherwise, the application scope should be limited to tabular data or metadata in a tabular format from fields like computer vision or natural language processing.
\
In line 126, the author states that NasBench 201 is a neural architecture search (NAS) method for computer vision (Dong & Yang, 2020). However, I think the Neural Ensembler is actually trained on tabular data, as indicated in the code at SearchingOptimalEnsembles/metadatasets/nasbench201/metadataset.py, lines 93-95. The author can compare the following two different pipelines:
- The author selects some base models and computes their predictions for each example. These predictions are then stored in tabular data format as examples to generate the training set. Lastly, train the Neural Ensembler on this set.
- Freeze all base models. Then, sample a mini-batch of examples from the raw training set. Lastly, Train the Neural Ensembler to generate the weights.

I believe the second schema is worth considering because 1) the input is raw data from the dataset, not the predictions from base models, 2) data augmentation can be used to avoid overfitting, and 3) for some recognition tasks, such as object detection, the predictions of different base models cannot be aligned, making it impossible to store base models' predictions in a tabular format.

I guess that the author may have used too many base models to construct the Neural Ensembler, which made it hard to train in the second mode with limited computing resources. This is just my guess, so I provide several solutions in the questions section for the author to refute this weakness.

**Questions:**

Overall, this paper requires some clarifications on theory and experiments. Given these clarifications, if provided in an author's response, I would consider increasing the score.

---

For the experiments, `any of the following` should be addressed.

1. We note that the authors compare the Neural Ensembler to a single model (see **lines 321-322**). It would have been better also to explore the efficiency of the Neural Ensembler, particularly considering the **number of model parameters** or **FLOPS** (see [1]). Otherwise, model scaling is a better choice for improving model performance. For example, we have a list of models:

| Model    | # Params | Acc@1  |
| -------- | -------- | -------|
| VGG19    | 143.7M   | 72.376 |
| ResNet18 | 11.7M    | 69.758 |
| ResNet34 | 21.8M    | 73.314 |
| ResNet50 | 25.6M    | 76.13  |
| ViT-B16  | 86.6M    | 81.072 |

- If ResNet18 and ResNet34 are used as base models for the Neural Ensembler, the Neural Ensembler’s performance should exceed that of ResNet50 due to its higher parameter count of over 33.5M (11.7M + 21.8M).
- When comparing with the VGG16 baseline, we should avoid using ResNet18, ResNet34, ResNet50, or ViT-B16 as base models for the Neural Ensembler, as each has better performance and a smaller scale.
- To demonstrate the advantages of the Neural Ensembler in computer vision and natural language processing, the author should use corresponding neural networks as base models and conduct experiments on image or text datasets.

2. It's important to note that the paper focuses on post-hoc ensembling of pre-trained models, rather than training base models from scratch. Thus, the author can clarify how their method compares the computational cost to other post-hoc ensembling methods.


[1] Shazeer, N., Mirhoseini, A., Maziarz, K., Davis, A., Le, Q., Hinton, G., & Dean, J. (2017). Outrageously large neural networks: The sparsely-gated mixture-of-experts layer. arXiv preprint arXiv:1701.06538.

---

> ### Author Response · Authors · 2024-11-21
> **Response to reviewer QKaE**
>
> Dear reviewer,
> thank you for your valuable feedback and insightful comments. We appreciate the opportunity to address your concerns and clarify aspects of our work.
>
> **Weaknesses**
>
> 1. ***The study lacks consideration of computational efficiency. The Neural Ensemblers’ performance depends on a strong subset of 50 base models, selected by DivBO, which may increase computational requirements.***
>
> We apologize for the misunderstanding. In reality, our Neural Ensembler does not depend on a strong subset of base models, we discuss this in **L431**. As demonstrated in our experiments addressing **RQ2**, our method can effectively ensemble base models selected at random.
>
> To address the computational efficiency, we included a comprehensive analysis in **Appendix F (Figure 9)** of the updated manuscript (or [here](https://anonymous.4open.science/r/NeuralEnsemblers/anonymous_rebuttal/runtime_vs_performance.pdf)). This analysis compares runtime costs of different post-hoc ensembling methods, including both training and inference times. The results indicate that our method exhibits lower runtime than other high-performing baselines, demonstrating that our **Neural Ensembler is both effective and computationally efficient, even without relying on a subset of strong base models**.
>
> 2. ***My review of the author's code confirmed that the meta-datasets are limited to a tabular format, making the paper's description somewhat vague. A clearer explanation of the data structure would have been beneficial. Otherwise, the application scope should be limited to tabular data or metadata in a tabular format from fields like computer vision or natural language processing. [...]***
>
> Thank you for bringing this to our attention. While we store base model predictions in tabular form for computational efficiency, our method is **not limited to tabular input data**. We included experiments on various modalities, including computer vision (e.g. *QuickTune*, *NASBench*) and natural language processing datasets (*FTC*). Storing the base model predictions as tables simply allows us to simulate the ensembles faster.
>
> Consider a regression task with two samples and three base models. The predictions from the base models are:
>
> $$ Z=
> \begin{bmatrix}
> z_{1,1}, z_{1,2}, z_{1,3} \newline
> z_{2,1}, z_{2,2}, z_{2,3}
> \end{bmatrix}
> $$
>
> Here, $z_{i,j}$ is the prediction of the j-th base model on the i-th sample. These predictions could come from any modality - for example, images processed by ResNet or ViT models.
>
> We assign weights to the base models:
>
> $$ W = \begin{bmatrix}
> w_1 \newline
> w_2 \newline
> w_3
> \end{bmatrix}
> $$
>
> We can simulate any combination of weighted ensembles as we just need to do a matrix multiplication for computing the ensemble prediction Y.
>
> $$Y=ZW$$
>
> Notice that we are still working with vision models; this tabular abstraction of the predictions facilitates the evaluation of baselines.
> Previous works have applied this technique to explore AutoML approaches (e.g. [1)].
>
> [1] Salinas, D., & Erickson, N. (2023). TabRepo: A Large Scale Repository of Tabular Model Evaluations and its AutoML Applications. arXiv preprint arXiv:2311.02971.
>
> 3. ***The author selects some base models and computes their predictions for each example. These predictions are then stored in tabular data format as examples to generate the training set. Lastly, train the Neural Ensembler on this set.***
>
> Thank you for summarizing this approach. Indeed, this is precisely the methodology we employ in our work. We apologize for the confusion.

---

> > ### Author Response · Authors · 2024-11-21
> > **Response to reviewer QKaE [continuation]**
> >
> > 4. ***We note that the authors compare the Neural Ensembler to a single model (see lines 321-322). It would have been better also to explore the efficiency of the Neural Ensembler, particularly considering the number of model parameters or FLOPS (see [1]). Otherwise, model scaling is a better choice for improving model performance.***
> >
> > We appreciate your suggestion to explore the efficiency of our Neural Ensembler. In the updated manuscript, we included a runtime analysis in **Appendix F (Figure 9)** (or  [here](https://anonymous.4open.science/r/NeuralEnsemblers/anonymous_rebuttal/runtime_vs_performance.pdf)), where we compare our Neural Ensembler with other post-hoc ensembling methods. This analysis includes both training and inference times, demonstrating that our method provides a favorable trade-off between performance and computational cost, even when accounting for the computational complexity.
> >
> > While scaling up individual models is indeed a viable strategy for improving performance, we believe that model scaling and ensembling are two **orthogonal approaches** that can **complement** each other rather than being mutually exclusive. Scaling models may increase performance, but it does not invalidate the benefits of ensembling methods. In fact, this is an actively researched area in the literature of mixture of experts (e.g. See related work section 5).
> >
> > Additionally, in the context of tabular data, large models are not always the optimal choice. Empirical evidence from competitions like Kaggle or software libraries for tabular data [2] shows that ensemble models often outperform single large models.
> >
> > Finally, ensembling offers practical benefits in terms of computational resources. For instance, if we have multiple GPUs with limited memory, we can train several smaller models in parallel and then ensemble their predictions. This approach can be more efficient than training a single large model, which might require complex model parallelism and incur communication overhead [3].
> >
> > [2] Erickson, Nick, et al. "Autogluon-tabular: Robust and accurate automl for structured data." arXiv preprint arXiv:2003.06505 (2020).
> >
> > [3] Brakel, Felix, Uraz Odyurt, and Ana-Lucia Varbanescu. "Model Parallelism on Distributed Infrastructure: A Literature Review from Theory to LLM Case-Studies." arXiv preprint arXiv:2403.03699 (2024).
> >
> > 5. ***To demonstrate the advantages of the Neural Ensembler in computer vision and natural language processing, the author should use corresponding neural networks as base models and conduct experiments on image or text datasets.***
> >
> > Thank you for this recommendation. We would like to highlight that we have conducted experiments using neural networks as base models on image datasets (*QuickTune*, *NASBench*) and text datasets (*FTC*), as suggested.
> >
> > 6. ***Thus, the author can clarify how their method compares the computational cost to other post-hoc ensembling methods.***
> >
> > As per your suggestion, we included a detailed comparison of the computational costs of our method and other post-hoc ensembling methods in **Appendix F (Figure 9)** of the updated manuscript (or [here](https://anonymous.4open.science/r/NeuralEnsemblers/anonymous_rebuttal/runtime_vs_performance.pdf)).
> >
> > 7. ***Using raw data for ensemble, instead of model predictions.***
> >
> > We appreciate your suggestion to explore using raw data for ensembling instead of model predictions. We discuss this option in **Appendix M** of the updated manuscript. Using raw data directly introduces challenges in generalizing the ensemble architecture across different modalities, as we would need to design a specific neural ensembler network for each type of data (e.g., ResNet or ViT for vision tasks). By using base model predictions, our approach remains **modality-agnostic** while still achieving top performance. Additionally, in **Appendix M (Figure 15)** we further show that, particularly for tabular data, using the raw input can lead to overfitting, thus adding complexity without significant benefits.
> >
> > ----
> > Finally, thank you very much for taking the time to review our paper and providing us with valuable feedback, prompting several improvements to our submission. We hope we have clarified some of your concerns and would appreciate it if you consider improving the rating in your review. We welcome any further suggestions or feedback that you would be willing to share.

---

> ### Comment · Reviewer_QKaE · 2024-11-25
>
> Thank you for providing further clarification. Based on your response, I have summarized my understanding of your work as follows:
>
> (1). Based on Appendix F (Figure 9), Neural Ensemblers outperform each single model but do not demonstrate computational efficiency advantages, which seems to contradict the authors' claims.
>
> (2). As per your explanation, the Dynamic Post-Hoc Neural Ensemblers cannot be considered end-to-end models. In the data pipeline, predictions from 50 models are first collected and saved as tabular input data, which is then processed by the Ensemblers.
>
> (4). To strengthen the argument, I recommend conducting quantitative ablation studies. Without considering the total number of model parameters or FLOPS, it becomes difficult to argue that multiple smaller models outperform a single large model. Even in scenarios with multiple GPUs and limited memory, a single large model can still be executed efficiently using Tensor Parallel (TP) or Pipeline Parallel (PP) techniques.
>
> (6). Similar to point 3, the response does not fully address the concern about computational efficiency in comparison to single large models.
>
> (7). The author's response, "Additionally, in Appendix M (Figure 15) we further show that, particularly for tabular data, using the raw input can lead to overfitting, thus adding complexity without significant benefits." is confusing. If the Ensemblers were run in an end-to-end mode without saving/loading tabular data file, the final output should not differ. Could you clarify why this approach would lead to overfitting?

---

> > ### Author Response · Authors · 2024-11-26
> > **Reply to reviewer**
> >
> > Thank you for the follow-up on our discussion.
> >
> > > ***(1). Based on Appendix F (Figure 9), Neural Ensemblers outperform each single model but do not demonstrate computational efficiency advantages, which seems to contradict the authors' claims.***
> >
> > Our work is around post-hoc ensembling, it is we assume that we have a set of fitted models.  We presented the post-hoc ensembling time i.e. time fitting the ensemble, not fitting the base models. Single best just takes the best model using the validation metric, which obviously is much faster than fitting an additional model. Notice that our method has better post-hoc ensembling time than competitive baselines such as greedy.
> >
> > > ***(2). As per your explanation, the Dynamic Post-Hoc Neural Ensemblers cannot be considered end-to-end models. In the data pipeline, predictions from 50 models are first collected and saved as tabular input data, which is then processed by the Ensemblers.***
> >
> > Indeed, as explained in the point above, we are doing post-hoc ensembling, and we are not training the model end-to-end. This has the advantage that we can ensemble models whose derivatives are difficult to compute e.g. XGboost, Random Forest, KNN, for tabular data.
> >
> > > ***(4). To strengthen the argument, I recommend conducting quantitative ablation studies. Without considering the total number of model parameters or FLOPS, it becomes difficult to argue that multiple smaller models outperform a single large model. Even in scenarios with multiple GPUs and limited memory, a single large model can still be executed efficiently using Tensor Parallel (TP) or Pipeline Parallel (PP) techniques.***
> >
> > Previous work showcases that an ensemble of models is better than large models in Tabular data [1]. Although this might not be the case for all the modalities, an ensemble has interesting properties over a single model [2]:
> >
> > * Performance-wise: an ensemble reduces the variance component of the prediction error by adding bias (avoiding overfitting); a large model cannot do this, thus it might overfit.
> > * Robustness-wise: an ensemble reduces reliance on any single model's predictions, making it better equipped to handle noisy data (e.g. outliers).
> >
> > We acknowledge that having a large model might be beneficial when having a lot of data, but this is not the general case. That being said, we want to highlight that comparing the efficiency of single models and ensembles is complementary to our work. If the base models in the ensemble are smaller or sparse, the ensemble can even be more efficient than the single dense model (as shown in https://arxiv.org/pdf/2202.11782, Table 2). Nevertheless, our method is agnostic to such choices, i.e. the pool of base learners that the Neural Ensembler constructs the ensemble from can contain dense, sparse or a mix of networks with different sizes.
> >
> > [1] Erickson, Nick, et al. "Autogluon-tabular: Robust and accurate automl for structured data." arXiv preprint arXiv:2003.06505 (2020).
> >
> > [2] Discussion forum from kaggle: https://www.kaggle.com/code/yeemeitsang/unreasonably-effective-ensemble-learning
> >
> > > ***(6). Similar to point 3, the response does not fully address the concern about computational efficiency in comparison to single large models.***
> >
> > Please refer to our previous response, in bullet point (1).
> >
> > > ***(7). The author's response, "Additionally, in Appendix M (Figure 15) we further show that, particularly for tabular data, using the raw input can lead to overfitting, thus adding complexity without significant benefits." is confusing. If the Ensemblers were run in an end-to-end mode without saving/loading tabular data file, the final output should not differ. Could you clarify why this approach would lead to overfitting?***
> >
> > The ensembles were indeed not trained end-to-end. Notice that not having end-to-end models is not the only way to overfit. We can also overfit by just making all models and the ensemble very good to this dataset. In the experiment in appendix M: 1) we first train models in the merged split, 2) then we built the ensemble using the same split. The resulting ensemble fits very well with this merged split, but does not generalize.
> >
> > ---
> >
> > Finally, we thank the reviewer for the effort in the review process.  We will incorporate the feedback in our future version.

---

### Official Review · Reviewer_Svxb · 2024-11-10

**Soundness:** 2
**Presentation:** 2
**Contribution:** 2
**Rating:** 3
**Confidence:** 3

**Summary:**

This paper introduces "Neural Ensemblers," a novel approach that uses neural networks to dynamically weight ensemble members for each input instance. Authors propose regularization technique of randomly dropping base model predictions during training.

**Strengths:**

- Paper is well written.
- Experimental results span different modalities

**Weaknesses:**

- Benefit of the proposed method is not super clear in Table 2. Given the variance of different methods, efficacy of the proposed method is unclear

- POC example given in the paper can be misleading as their dynamic ensemblers train more parameters as compared to baseline. It remains unclear if this would be a problem in the first place if we could train base models that are sufficiently overparameterized

- It remains unclear what are the main reasons due to which the proposed methods improves on top of simple ensembling baselines.

- Post-hoc ensembling in the name can be misleading for someone who is not aware of the literature. In particular, to obtain the ensemble we need to train the parameters of the ensemble model.

I am not very well aware of the literature to comment on the novelty of paper, so I will leave that on other reviewers and the area chair.

**Questions:**

- It is not clear when dynamic ensemblers would benefit? Is it the case that dynamic ensemblers improve because base models underfit the data? Does benefit of these ensemblers persist in cases when the base models are over parameterized neural networks?

- More data is needed to train the parameters of the dynamic ensemblers  It is unclear if we can reuse the training data used for base models or do we need more training data? In case we need more training data, a favorable comparison would be using that data in the first place for base models and perform standard ensembling techniques.

---

> ### Author Response · Authors · 2024-11-21
> **Response to reviewer Svxb**
>
> Dear Reviewer,
> thank you for your valuable feedback and insightful comments. We have addressed your concerns below:
>
> **Weakneses**
>
> 1. ***The benefit of the proposed method is not super clear in Table 2. Given the variance of different methods, the efficacy of the proposed method is unclear. [...]***
>
> Thank you for your valuable feedback and for pointing out the need for a clearer demonstration of our method's benefits in **Table 2**. We understand that, given the variance among different methods, it might not be immediately evident how our method stands out in terms of efficacy. We want to clarify the benefits by pointing out the following aspects of our work:
>
> > * Comprehensive Performance Metrics: In the original manuscript, **Table 2** and **Table 3** reported various metrics: the average normalized error, AUC, and average normalized Negative Log-Likelihood (NLL), showing that our method consistently achieves **top performance** across various datasets and modalities. A more detailed list of results can be found in **Appendix E**.
>
> > * New Critical Difference (CD) Diagrams: To add a layer of statistical interpretability, we computed Critical Difference diagrams, which visualize the performance differences among methods while considering **statistical significance**. As shown in **Figure 11** of the updated manuscript (or [here](https://anonymous.4open.science/r/NeuralEnsemblers/anonymous_rebuttal/CD-Difference.pdf)), our method achieves the best average rank and is significantly better than many baselines. Additionally, statistically, no other method is better than ours.
>
> > * New Computational Efficiency Analysis: Recognizing that practical efficacy also depends on computational cost, we analyzed the runtime performance of our method compared to others. **Figure 9** in the updated manuscript (or [here](https://anonymous.4open.science/r/NeuralEnsemblers/anonymous_rebuttal/runtime_vs_performance.pdf)) illustrates that our method not only **excels in performance but also is faster than several baselines**.
>
> > * New Insights about Significance on Datasets with a lot of Classes: In section N (figure 17) we demonstrate that our method is statistically significantly better than the second best method (Greedy) in datasets with a lot of classes. The reviewer can also see the results [here](https://anonymous.4open.science/r/NeuralEnsemblers/anonymous_rebuttal/aggregated_across_metadatasets_with_a_lot_of_classes.pdf). We hypothesize this occurs because of the neural ensembler's design, as it shares parameters across different classes (section 3.1).
>
> > * Theoretical Guarantees: To support our empirical findings, we provide **theoretical proofs**, ensuring that our method promotes ensemble diversity, which is crucial for limiting overfitting and enhancing generalization. These theoretical contributions are detailed in **Propositions 1, 2, and 3** of our paper. Our ablation study in Figure 6 configure that using the base models DropOut is indeed very helpful, i.e. DropOut rate >0 is yields the best performance across many datasets.
>
> 2. ***POC example given in the paper can be misleading as their dynamic ensemblers train more parameters as compared to baseline. It remains unclear if this would be a problem in the first place if we could train base models that are sufficiently overparameterized. Does benefit of these ensemblers persist in cases when the base models are over parameterized neural networks?[...]***
>
> Thank you for raising this important point. To address your concern, we extended our proof-of-concept experiment to test whether the benefits of our dynamic ensemblers persist with **overparameterized base models**.
>
> We increased the complexity of the base models by using polynomials of degree 10 instead of degree 2. The results, shown in **Appendix G (Figure 10)** (or [here](https://anonymous.4open.science/r/NeuralEnsemblers/anonymous_rebuttal/motivation_ensemble_overparm_base_models.pdf)), demonstrate that even with these more expressive base models, our dynamic ensemblers continue to outperform the baselines.
>
> This indicates that the advantages of our method are not limited to scenarios with underparameterized base models. Even when base models are sufficiently overparameterized, our dynamic ensemblers provide improved performance, due to their **ability to mitigate overfitting** and **adaptively combine model predictions**.

---

> ### Author Response · Authors · 2024-11-21
> **Response to reviewer Svxb [continuation]**
>
> 3. ***It remains unclear what are the main reasons due to which the proposed methods improves on top of simple ensembling baselines. [...]***
>
> Thank you for your thoughtful comment. We'd like to clarify the main reasons why our proposed method outperforms simple ensembling baselines. Our method introduces two key innovations that collectively enhance performance.
>
> **Dynamic, Instance-Dependent Ensembling**: Unlike traditional ensemble methods that assign fixed weights to base models, our approach computes ensemble weights dynamically for each input instance. This means that for each data point, the ensemble can emphasize the base models that are most relevant to that specific instance. This **adaptive weighting** allows the model to tailor its predictions based on the unique characteristics of each input, leading to improved accuracy.
>
> **Regularization through Dropout and Parameter Sharing: During training, we apply dropout to the predictions of the base models**. This serves as a regularization technique, promoting diversity among the ensemble members and preventing the model from overfitting to the training data. Additionally, our Neural Ensembler employs a **shared neural architecture** with parameter sharing across different components. This reduces the total number of parameters and encourages the model to learn more generalizable features.
>
> No prior work or baseline considers these aspects jointly. Traditional ensembling methods typically use fixed weights and do not incorporate instance-dependent weighting or these specific regularization strategies.
>
> 4. ***Post-hoc ensembling in the name can be misleading for someone who is not aware of the literature. In particular, to obtain the ensemble we need to train the parameters of the ensemble model.***
>
> Thank you for bringing this to our attention. We understand that the term "post-hoc ensembling" may be unfamiliar to some readers and could potentially cause confusion. In our work, "post-hoc ensembling" refers to methods where the **base models are already trained**, and the ensemble model is focused on aggregating their predictions without retraining the base models themselves.
>
> While it is true that we need to train the parameters of the ensemble model, the key aspect of post-hoc ensembling is that it operates on pre-trained base models. This approach is discussed in previous works [1, 2], where the ensemble aims to enhance performance by optimally combining existing models.
>
> [1] Purucker, Lennart Oswald, and Joeran Beel. "CMA-ES for Post Hoc Ensembling in AutoML: A Great Success and Salvageable Failure." International Conference on Automated Machine Learning. PMLR, 2023.
>
> [2] Shen, Yu, et al. "DivBO: diversity-aware CASH for ensemble learning." Advances in Neural Information Processing Systems 35 (2022): 2958-2971.
>
> **Questions**
>
> 1. ***It is not clear when dynamic ensemblers would benefit? Is it the case that dynamic ensemblers improve because base models underfit the data?***
>
> Thank you for your question. Dynamic ensemblers benefit in situations where the optimal combination of base models varies across different instances, making instance-dependent aggregation advantageous. This means that each sample can be better predicted by weighting base models differently based on its specific characteristics. As highlighted in our motivation section (**Section 2**), dynamic ensembling allows the model to adapt to diverse patterns within the data.
>
> Importantly, this benefit is not limited to cases where base models underfit the data. Our experiments with overparameterized base models (*Appendix G (Figure 10)**) demonstrate that dynamic ensemblers continue to improve performance even when base models are sufficiently expressive. This suggests that the advantage of dynamic ensembling comes from its **ability to adapt predictions to individual instances**, enhancing performance beyond what is achievable with static ensemble methods.
>
> Finally, Quicktune [1] and Nasbench [2] metadatasets have been used in previous papers, demonstrating that they contain well-tuned models.
>
> [1] Arango, Sebastian Pineda, et al. "Quick-tune: Quickly learning which pretrained model to finetune and how." arXiv preprint arXiv:2306.03828 (2023).
>
> [2] Ying, Chris, et al. "Nas-bench-101: Towards reproducible neural architecture search." International conference on machine learning. PMLR, 2019.

---

> ### Author Response · Authors · 2024-11-21
> **Response to reviewer Svxb [continuation 2]**
>
> 2. ***More data is needed to train the parameters of the dynamic ensemblers It is unclear if we can reuse the training data used for base models or do we need more training data? [...]***
>
> Most of the baselines (except the *Random* baseline) use additional data (validation data) to optimize the ensemble. We run an experiment when we decrease the validation data size, and see that the performance of our method barely decreases in some metadatasets. For these experiments refer to **Appendix K (Figure 14)** (or  [here](https://anonymous.4open.science/r/NeuralEnsemblers/anonymous_rebuttal/pct_valid_data.pdf)).
>
> Additionally, we run an experiment where we train the base models and the ensemblers on the same data (merging the training and validation split). We can see the results in **Appendix L (Figures 15 and 16)** (or [here](https://anonymous.4open.science/r/NeuralEnsemblers/anonymous_rebuttal/figure15.png)), showing that training the base models and the ensemblers leads to overfitting. Thus it is better to train the ensemble in a separate split. This behavior is exhibited by other ensemble strategies such as *Greedy* baseline.
>
> ---
> Finally, thank you very much for taking the time to review our paper and providing us with valuable feedback, prompting several improvements to our submission. We hope we have clarified some of your concerns and would appreciate it if you consider improving the rating in your review. We welcome any further suggestions or feedback that you would be willing to share.

---

### Author Response · Authors · 2024-11-21
**General Response**

We thank all the reviewers for their insightful feedback. We are happy that the reviewers consider our work novel, with a significant contribution to the field. As a general response, we listed here the changes to the submission PDF (highlighted in blue), including results after running experiments suggested by the reviewers:

* Extended discussion on related work: **Section 5**, as a response to reviewer  **eck1**.
* Unnormalized metrics results: **Appendix E1**, as a response to reviewer  **62Pt**.
* Analysis of computation cost: **Appendix F**, as a response to reviewers **Svxb, QKaE**.
* Proof of concept with overparametrized base models: **Appendix G**, as a response to reviewer **Svxb**.
* Critical difference diagrams: **Appendix H**, as a response to reviewers **Svxb,  62Pt**.
* Results with additional baselines (Catboost, XGBoost, LightGBM, Akaike weighting): **Appendix I**, as a response to reviewer **eck1**.
* Sensitivity analysis of neural ensemble hyperparameters: **Appendix J**,  as a response to reviewers **62Pt, eck1**.
* Ablation study of validation data size: **Appendix K**, as a response to reviewers **Svxb,  62Pt**.
* Effect of merging training data: **Appendix L**, as a response to reviewer **Svxb**.
* Results of Neural Ensembles operation on the original input space: **Appendix M**, as a response to reviewer **Svxb**.
* Insights of the significance in datasets with a lot of classes: **Appendix N**, as response to reviewer **Svxb, 62Pt**.

We believe our changes address the reviewers' concerns and reinforce the value of our work. Our replies further strengthen the validity of our research hypothesis. We also respond to each reviewer individually below.

---

### Note · Authors · 2024-11-26

**Comment:**

We thank the reviewers for their time and feedback. We have decided to withdraw our submission for now and integrate the points from this discussion before submitting it to the next conference.

**Withdrawal Confirmation:**

I have read and agree with the venue's withdrawal policy on behalf of myself and my co-authors.